# Chromogranin A: An Endocrine Factor of Pregnancy

**DOI:** 10.3390/ijms24054986

**Published:** 2023-03-05

**Authors:** Michalina Bralewska, Tadeusz Pietrucha, Agata Sakowicz

**Affiliations:** Department of Medical Biotechnology, Medical University of Lodz, Zeligowskiego 7/9, 90-752 Lodz, Poland

**Keywords:** chromogranin A, pregnancy, catestatin

## Abstract

Pregnancy is a state of physiological and hormonal changes. One of the endocrine factors involved in these processes is chromogranin A, an acidic protein produced, among others, by the placenta. Although it has been previously linked to pregnancy, no existing articles have ever managed to clarify the role of this protein regarding this subject. Therefore, the aim of the present study is to gather knowledge of chromogranin A’s function with reference to gestation and parturition, clarify elusive information, and, most importantly, to formulate hypotheses for the future studies to verify.

## 1. Introduction

During pregnancy, the maternal organism must undergo multiple physiologic adaptations to meet the demands of the developing fetus. In the center of the ongoing changes is the placenta, an endocrine organ present in pregnancy. Placental hormones support maintenance of gestation, enable fetal growth, and are involved in the mechanisms of parturition.Among the well-known and widely studied hormones of pregnancy are progesterone, placental lactogen (hPL), corticotropin-releasing hormone (CRH) and human chorionic gonadotropin (hCG) [1]. However, there exist endocrine factors whose role in pregnancy remains elusive. One of them is chromogranin A (CgA), an acidic protein mainly localized within chromaffin granules of the adrenal medulla and in sympathetic nerves, detected for the first time in the placenta in 1996 [2,3]. So far, several possible roles of CgA and its derived peptides have been proposed, strongly emphasizing their influence on the endocrine system. Yet, except for single reports, none of those roles have ever been attributed directly to CgA in pregnancy. Available data allow us to assume that CgA itself, as well as some of the peptides generated from its cleavage, are engaged in apoptosis, immune response, or blood pressure regulation, among others [4,5,6]. Further, in hypertensive pheochromacytoma patients, CgA concentrations are used as diagnostic markers, and only tumor resection is known to be a cure [7,8]. A similar phenomenon is observed in pregnancy-related disorders, such as in preeclampsia (PE), where hypertension is one of the main characteristics and placenta removal (source of CgA) constitutes the only mode of treatment [9]. Therefore, the aim of the present study is to gather knowledge of CgA’s function with reference to gestation and parturition, to clarify elusive information scattered across the existing articles, and, most importantly, to formulate hypotheses for future studies to verify.

## 2. Hormonal Balance of Pregnancy

Gestation is a state of extreme hormonal fluctuations. The majority of the hormones are secreted in order to establish and maintain a healthy pregnancy. Among most prominent gestation guardians are progesterone, estrogens, and hCG. The roles of these and other fundamental hormones of gestation are summarized in Table 1.

Estrogens are a group of four steroid hormones (estrone, 17β-estradiol, estriol, and estetrol), the most abundant one being estradiol. It promotes embryo implantation, helps to regulate uteroplacental blood flow by promoting angiogenesis and vasodilatation, and is involved in labor initiation [10]. Progesterone, as well as estrogen, is a steroid hormone mainly produced by the corpus luteum. Elevated progesterone levels have been shown to prevent pregnancy loss and preterm labor. This effect is, at least in part, mediated by the influence on cytokine production. Progesterone has been shown to inhibit Th1 (T helper 1) cytokines and, at the same time, to stimulate production of Th2, a pregnancy-protective cytokine. Furthermore, it is also engaged in the preparation of the endometrium for implantation and regulation of the trophoblast’s invasion and migration [11,12]. 

Another hormone among most important hormones of pregnancy is human chorionic gonadotropin. Its critical role is related to progesterone and estrogen production regulation, as it protects the corpus luteum from involution [1]. Furthermore, hCG can modulate secretion of trophoblastic matrix metalloproteinases (MMPs) and endometrial tissue inhibitors of metalloproteinases (TIMPs), which indicate hCG’s role in trophoblast invasion [13]. It has also been reported that hCG may have a positive influence on immunotolerance and placental angiogenesis [10].

However, there are many other hormones in pregnancy (hPL, relaxin, prolactin, cortisol, etc.), which are relevant to the process of pregnancy establishment, maintenance, and later labor. Molecules of unknown function also exist. One such molecule is the chromogranin A hormone. Since hormonal balance is a key factor in pregnancy prosperity, it is essential to fully scrutinize the function of each hormone, with the main focus on the least recognized ones. 

**Table 1 ijms-24-04986-t001:** Fundamental hormones of pregnancy and their role [1,10,14].

Hormone Name	Mainsource in Pregnancy	Role in Pregnancy
Progesterone	Corpus luteum Placenta	Prevention of preterm labor Prevention of maternal rejection of the fetus Guidance for successful implantation Enhancement of the expression of Th2 cytokines Preparation of the mammary gland for lactation
Estrogens	Corpus luteum Placenta	Increase in and regulation of the uteroplacental bloodflow Promotion of embryo implantation by endometrial growth and differentiation stimulation Enhancement of the syncytialization of human primary cytotrophoblast
Human placental lactogen (hPL)	Placenta (syncytiotrophoblast)	Rise in maternal plasma insulin-likegrowth factor-1 (IGF-1) concentrations Regulation of maternal lipid and carbohydrate metabolism
Human chorionic gonadotropin (hCG)	Placenta (trophoblast)	Rescue of the corpus luteum from involution Enhancement of the progesterone synthesis by the corpus luteum Promotion of invitro human cytotrophoblast differentiation into the syncytiotrophoblast Stimulation of the trophoblast invasion
Relaxin	Corpus luteum	Promotion of decidualization Modulation of matrix metalloproteinase activity Important endometrial/decidual angiogenic factor
Placental growth hormone (HPGH)	Placenta (syncytiotrophoblast)	Induction of insulin resistance Promotion of gluconeogenesis, lipolysis, and anabolism Increasing the nutrient availability for fetal nourishment
Leptin	Placenta (syncytiotrophoblast)	Invitro stimulation of theproliferation of cytotrophoblast cells Involvement in the trophoblast invasion, placentation, and implantation

## 3. Chromogranin A Family and Structure

Granins belong to the family of acidic proteins produced by endocrine, neuroendocrine, and neuronal cells. This family comprises 9 granin proteins: chromogranin A, B, and C (secretogranin II), as well assecretogranin III, IV, V, VI, VGF, and proSAAS. The first granin to be discovered and, at the same time, the most widely studied is chromogranin A (CgA) [15,16]. Encoded by the CHGA gene, located on chromosome 14, human CgA weighs 48 kDa and is composed of 439 amino acids [17,18]. Thanks to its multiple cleavage sites, CgA can be processed by proteolytic enzymes such as prohormone convertases 1 and 2 (PC1 and PC2), furin, or plasmin [19,20]. As a result, a series of different biologically active peptides can be obtained (Figure 1). Some of them exert similar effects, while others exhibit completely opposite responses, e.g., the negative inotropic effect of CST vs. the positive inotropic effect of serpinin (Table 2). Therefore, regulation of CgA processing in order to produce the proper molecules seems to be crucial for homeostasis maintenance. The most significant ones, in terms of the discussed issue, are vasostatin (VS-I), catestatin (CST), and serpinin.

The N-terminal CgA fragment vasostatin I (CgA_1-76_; VS-I) exhibits a wide range of biological activities. Generally, it is known for its antimicrobial and antifungal effects, its influence on vasculature and vessel tension, and its modulation of cell adhesion [18]. Catestatin (CgA_352-372_; CST), on the other hand, is a C-terminal CgA peptide with vasodilatory effectsthat inhibits catecholamine secretion through action at nictotinic cholinergic receptors [10]. The last one described, serpinin, was identified as a 2.9-kDa C-terminal CgA peptide. Serpinin plays a crucial role in granule biogenesis through up-regulation of the protease inhibitor protease nexin-1 (PN-1), preventing granule degradation [22]. 

Thus far, a little is known about CgA’s influence on hormonal homeostasis. Although CgAwas identified in the placenta for the first time in 1996 [3], its role in pregnancy has not been elucidated, and the available literature only vaguely answers some of the emerging questions. It is only known that the level of CgA increases with the course of gestation. It is believed that this protein is secreted from trophoblastic cells both into maternal circulation and into the amniotic fluid, where its levels are significantly higher at the termination of gestation (i.e., weeks 37–42 of gestation) than in the early second trimester of gestation (i.e., weeks 14–17). This might indicate that CgA plays an important role in the preparation of maternal tissue for the delivery. This hypothesis was confirmed by other studies indicating that CgA levels are significantly higher in the umbilical blood and plasma of children delivered vaginally than in those delivered by caesarean section [3,23]. Interestingly, the CgA derivative peptide pancreastatin (PST-LI) presents reverse association with the time of gestation; the levels of PST-LI are significantly higher at the beginning of gestation than at term [3]. This might indicate that the CgA undergoes various cleavages into its derivative peptides at different stages of gestation, allowing one protein to determine the success of the pregnancy. Therefore, the beginning of gestation might be related to the secretion of peptides implicated in the activation of inflammatory reactions and the induction of angiogenesis. The middle of gestation might depend on the peptides supporting the maternal immunosuppression, whereas at the end of gestation, the cleavage of CgA is silenced, and the whole protein might regulate the preparation of the maternal organism for partition by a mechanism that is currently unknown. To date, few hypothetical functions or influences of CgA and its derived peptides on the course of pregnancy are known (Figure 2). Similarly, CgA’s influence on the hormonal balances of pregnant women is highly probable, and needs to be investigated to dissipate doubts arising from insufficient knowledge. 

**Table 2 ijms-24-04986-t002:** CgA-derived peptides and their most important biological functions.

CgA-Derived Peptide	Position	Biological Function(s)
Chromostatin	CgA_1-20_	Involvement in Pancreatic β-cell functions [18,24]
Vasostatin I and II (VS-I and VS-II)	VS-I: CgA_176_ VS-II: CgA_1-113_	Vasodilation Antifungal and antimicrobial effects Modulation of cell adhesion Inhibition of parathyroid hormone secretion Regulation of cardiovascular system Inhibition of vessel permeability Negative inotropic effect [18,25,26]
Chromofungin (CHR)	CgA_47-66_	Anti-inflammatory and anti-microbial properties [27]
Chromacin	CgA_173-194_	Antimicrobial effects [28]
Pancreastatin (PST)	CgA_250-301_	Glucose and lipid metabolism [18]
Catestatin (CST)	CgA_352-372_	Vasorelaxant and antihypertensive effects Angiogenic and vasculogenic activities Cardiovascular system regulation Inhibition of catecholamine secretion Negative inotropic effect [18,25,29]
WE14	CgA_324-337_	Modulation of mast cell histamine secretion [30]
GE25	CgA_347-365_	-
Parastatin	CgA_357-428_	Modulation of parathormone release [18]
Serpinin	CgA_402-439_	Granule biogenesis β-adrenergic inotropic and lusitropic modulator [22,31]

## 4. Crucial Pregnancy Adaptations: Chromogranin A and CgA-Derived Peptides as Endocrine Factors of Pregnancy

### 4.1. Cardiovascular Adaptations in Pregnancy

As mentioned previously, the maternal organism needs to prepare itself for the demands of the growing fetus. Changes to the cardiovascular system occur in early pregnancy. There is an increase in cardiac output, probably primarily caused by endothelium-dependent factors such as vasodilatory prostaglandins (e.g., PGI2) or up-regulation of nitric oxide (NO) synthesis [32]. Peripheral vasodilatation leads to a decrease in systemic vascular resistance and, as compensation, cardiac output increases. The dominant evidence shows that blood pressure (systolic, diastolic, and mean arterial pressure) decreases during pregnancy. However, recent studies have contradicted that theory, indicating a drop in blood pressure at the beginning of pregnancy, followed by a progressive increase over the course of gestation [33,34].

Another of the most critical adaptations in pregnancy is vascular remodeling. This process, performed to fulfill the fundamental requirements of the fetus, depends on multiple cellular processes, such as migration, proliferation, death, and production/degradation of the extracellular matrix. Before pregnancy, spiral arteries are characterized by muscular walls and elastic lamina. During gestation, these highly-resistant, low-inflow vessels progressively lose muscle layers in favor of extravillous trophoblast (EVT) cells.Furthermore, the diameter of the spiral arteries increases by at least 10 times, enabling better blood supply to the fetus with reduced pressure [35,36]. Maternal vascular remodeling with EVT invasion through decidual tissue ensures appropriate placental perfusion and contributes to establishment of a successful pregnancy. This process is constantly being controlled by multiple factors which can alter ongoing changes. Among those factors are chemokines, cell adhesion molecules, cytokines, growth factors, extracellular matrix enzymes, environmental oxygen, and decidual immune cells [36,37], most of which cooperate with the vascular endothelial growth factor (VEGF). 

Mechanisms by which VEGF acts involve activation of mitogen-activated protein kinases (MAPKs) and extracellular signal-regulated kinases (ERKs). According to the literature, decidual angiogenesis and vascular remodeling are mainly regulated through VEGF-A [38]. In vitro experiments revealed that treatment of human umbilical endothelial cells (HUVEC) with CgA-derived VS-I results in the inhibition of VEGF-induced ERK phosphorylation, migration, proliferation, morphogenesis, and invasion. Additionally, VS-I was suggested to inhibit the formation of capillary-like structures [26]. Further studies confirmed the proposed theory, proving VS-I’s anti-angiogenic properties [39]. Some of these studies pointed that VS-I is a strong regulator of the translocation of hypoxia-induced factor 1 alpha (i.e., Hif-1α) from the cytoplasm into the nucleus [39]. This factor is recognized as a key player in the induction of angiogenesis; thus, the inhibition of its migration into the nucleus by VS-I might down-regulate the process of placentation. Interestingly, another CgA-derived peptide, CST, was proven to induce endothelial cell migration and proliferation, as well as to act as a pro-angiogenic factor [40]. Similarly to VEGF, CST regulates angiogenesis via the protein G and MAPK pathway. However, this peptide works independently from VEGF. The inhibition of VEGF does not influence CST-induced processes [41]. Furthermore, recent studies conducted on maternal serum from women with PE have suggested that CST levels may be associated with fetal cardiovascular dysfunction [42] In this cross-sectional study, women were divided into two study groups according to the onset of PE (early or late). According to the outcomes of an enzyme-linked immunosorbent assay (ELISA), patients in the early-onset group were characterized by lower levels of CST in maternal serum as compared to the healthy controls. However, the main limitation of the presented study was the small group size used for the experiment (27 and 28 for the early and late PE groups, respectively, and 28 for the control group). Taken together, these opposite modes of actions of both CgA peptides suggest that the final visible effect is probably dependent on specific CgA processing, which, in turn, may be reliant on locally active peptidases. Thus, depending on differing enzymatic cleavage behaviors, CgA may present either a damaging or protecting influence on the outcome of the pregnancy.

### 4.2. Renal Adaptations, Blood Pressure, and Vasomodulatory Effects during Preganancy

Basic pregnancy adaptations concern visible changes in renal structure, function, and hemodynamics. During pregnancy, the size of the kidneys increases (up to 30%), and, at the same time, there is an observable increase in renal plasma flow (RPF) and the glomerular filtration rate (GFR) [43]. These changes are controlled by multiple hormones, as well as the physical demands of the maternal organism. Among the main hormones engaged in the aforementioned processes are estrogens, progesterone, relaxin, and rennin-angiotensin–aldosterone system (RAAS). Moreover, the role of CgA, especially its amino terminus VS domain, is very well-described considering the regulation of renal function. The renal function declined together with the elevation of CgA concentration in the human plasma. In light of Chien et al.’s study, CgA-derived CgA_1-40_ peptide induced calcium-dependent secretion of all Weibel–Palade bodies and increased TGF-β-1 secretion from mesangial cells, influencing the depletion of GFR [44]. Therefore, it should be suspected that the Cg_1-40_ level has a tendency to deplete alongwith the development of pregnancy. This depletion might be regulated by sex hormone that is dominant during pregnancy. The study conducted by Fischer-Colbrie demonstrated that estrogen treatment significantly influences the down-regulation of the *CHGA* gene’s expression level in male rats. The dependence of the CgA level on estrogens has been confirmed in human and animal models other than rats [45].

The increased cardiac output and dilation of maternal vessels fulfill the demands of the developing fetus. One of the molecules engaged in these processes is NO, a biological mediator synthesized by nitric oxide synthetase (NOS). In addition to the aforementioned functions, NO is also implicated in pregnancy by regulating ovulation, implantation, labor, and delivery [46]. An in vivo test on mice revealed that targeted CgA gene ablation (*Chga-/-*) caused increased lipid peroxidation and higher reactive oxygen species (ROS) activity. Changes were marked by increased urinary isoprostane excretion and H_2_O_2_, respectively. Furthermore, *Chga-/-* mice displayed depletion in their NO levels, as well as phenotypic changes such as elevated blood pressure (BP) [4], suggesting a possible role of CgA (or one of its peptides) in the maintenance of homeostasis in pregnancy.

In terms of BP regulation, two CgA-derived peptides, VS-I and CST, must be mentioned. Both are known for their vasodilatory effects, although they originate from different mechanisms. In vitro tests on cardiomiocytes revealed that VS-I induces a Ca^2+^-independent/PI3K-dependent increase in endothelial NO production [47,48]. In the other hand, there exist some contradictory studies showing that VS-I may demonstrate distinct effects in distinct parts of the peripheral vasculature [49]. The vasodilatory action of the second peptide, CST, is, at least in part, mediated by histamine release and activation of histamine receptors (H1) [47]. Studies on humans showed that vasodilation caused by CST was dose-dependent, and female subjects exhibited grater vasodilation than male subjects. This fact can be explained by interesting finding: despite having lower *CHGA* precursor concentrations, female subjects had higher plasma CST levels [50]. In the light of the quoted results, it is interesting that, in our latest study conducted with placental tissue, the *CHGA* gene expression level was significantly higher in the PE group in comparison to the healthy controls. In the same time, CST protein levels were significantly diminished in the study group [51]. Therefore, it may be suggested that depletion of the CST level, which plays a protective role in hypertension development, might be a marker of developing PE. In contrast to our outcomes are studies conducted by Nevin Tüten et al., who found that CST levels in the blood of preeclamptic women (both with mild and severe preeclampsia) were higher than in the controls [52].This might suggest that the increase in the production of CST in PE is a maternal compensatory mechanism to try to maintain the blood tension balance. However, the authors note that in this study, the level of chromogranin (the main source of catestatin) was not tested. It is possible that in PE, the level of CgA increases, as was published elsewhere [51]; therefore, the levels of CgA cleavage products, which act as regulators of blood pressure, are higher in the serum of preeclamptic mothers than in the control population.This might explain the contrary results between studies; some of them pointed to the differences in CST levels between preeclamptic and non-complicated pregnancies [51,52], whereas other studies did not notice that relationship [42]. Furtherobserved differences between our previous study and the study of Tüten et al. come from the different inclusion/exclusion criteria, as the authors dividedthe patients into mild and severe PE groups, while in our previous study, the division was based on the time at which PE developed. Furthermore, in the inclusion criteria, Tüten N. gave additional disease entities, such as renal insufficiency, in which the renal disease itself is connected with increased plasma CST levels.

Therefore, further studies are necessary to test whether the ratio of CgA to its products, which decrease (i.e., catestatin) or elevate BP, differs between populations of preeclamptic and normotensive pregnant women.

### 4.3. Maternal Immune Tolerance Adaptations

Despite maintaining a proper environment for the developing fetus, maternal and fetal coexistence is also crucial for a successful pregnancy. The fetus, as a semi-allogenic entity, would normally be attacked by the immune system; therefore, rearrangement of the maternal defense system is crucial for gestation maintenance. One of the most prominent mechanisms of maternal–fetal tolerance is the Th1/Th2 immune shift. This immune switch results in a substitution of highly pro-inflammatory Th1 phenotype by an anti-inflammatory Th2 profile. Eventually, there is an increased production of protective interleukins: interleukin 10 (Il-10), Il-4, and Il-6, which in turn prevent fetal allograft rejection [53]. Th2 cytokines further promote the production of M2 macrophages, which, in contrast to the classically activated M1 type, are considered as anti-inflammatory and pregnancy-protecting factors [54]. 

Among other key players involved in pregnancy maintenance are uterine natural killer (uNK) cells. uNKs exhibit weak cytotoxic profiles; thus, they do not kill trophoblast cells.Furthermore, uNK cells produce multiple immunoregulatory cytokines, angiogenic factors, and matrix metalloproteinases, all of which are involved in pregnancy establishment by influencing angiogenesis, extracellular matrix remodeling, and trophoblast invasion [53] As the developing fetus is perceived by the maternal immune system as a semi- allogenic graft, to avoid rejection, the maternal immune response must switch into a non-invasive mode. Not only does the Th1/Th2 balance change, but multiple immune cells also need to be optimally regulated to prevent a detrimental response to the allogenic fetal cells. One of the T regulatory cell populations, the cluster of differentiation 8 (CD8^+^), is of a special importance concerning this matter. CD8^+^ T cells (cytotoxic T lymphocytes) are known for their contribution to the initiation, progression, and regulation of pathogenic autoimmune responses, releasing cytokines and chemokines. In the other hand, CD8^+^T cells can also down-regulate autoimmune responses [55]. Those opposite effects are likely possible thanks to the differently acting CD8^+^ T cell subpopulations, which have yet to be thoroughly examined. A study conducted by Yi Li et al. revealed that CgA can generate autoreactive CD8^+^ T cells. The studied CD8^+^ T cells population was assessed as proinflammatory, and was suggested to participate in the pathogenic process of the autoimmune disease type 1 diabetes [56]. The proinflammatory phenotype of the tested cells was determined upon their production of cytokines, such as IL-17, and interferon gamma (IFN-γ), both of which are connected with pregnancy. Depending on the circumstances, those cytokines can act in favor of or against pregnancy success, e.g., physiologically, they are involved in the remodeling processes of spiral arteries, while when overexpressed, IL-17 and IFN-γ can influence PE and abnormal brain development [57,58,59]. Since CgA has an impact on IL-17 and IFN-γ production by CD8^+^ T cells, it is necessary to define whether this influence is connected with physiologic, or, rather, pathologic processes of pregnancy. Interestingly, in terms of pregnancy maintenance, CD8^+^ T cells seem to have a protective influence. Shao et al. report that human placental trophoblasts activate a CD8^+^ T cell population whichregulates T cell-dependent B cell responses and suppresses Ig secretion, suggesting their role in the prevention of fetal rejection [60]. Additionally, an in vivo study conducted on mice showed that mice exposed to stress exhibited a decrease in the pregnancy-protective uterine CD8^+^ T cell subpopulation [61]. This decrease was able to be restored by a progesterone-derivative application known asdydrogesterone.

Progesterone is well-known for its protective effect on pregnancy. Its role is connected with the fact that the Th1/Th2 immune response shift present in pregnancy is controlled by an immunomodulatory protein known as progesterone-induced blocking factor (PIBF). PIBF is, among others, secreted by CD8^+^ T cells upon interaction of progesterone with progesterone receptors on their surface.Considering this fact, as well as the above results, it was suggested that the tested population of CD8^+^ T may have a crucial role in Th1/Th2 balance, determining the pregnancy’s success or failure [42]. Interestingly, some observations indicate that some CgA-derived peptides, e.g.,pancreastatin, might be implicated in adverse pregnancy outcomes. The inflammatory reaction in the adipose tissue of diabetic mice treated by a PST inhibitor (i.e., PSTi8) was significantly attenuated in comparison to the controls. The studied mice presented depletion in the expressions of both the TNF-α and IL-6 genes. Moreover, the polarization of classically activated macrophage M1 toward M2 was observed in PSTi8-treated animals [62]. This balance between M1 and M2 with a shift toward M2 macrophages also has a significant impact on the development of gestation.Maternal immunosuppression, as well as embryo implantation, placental formation, embryonic development, and delivery processes undergo regulation by the cytokine profile secreted by macrophages stationed in decidua tissue [63]. In pathological pregnancies, such as cases of miscarriage or PE, the macrophage fraction is skewed toward the M1 phenotype; thus, it might be suggested that overexpression of PST in pregnancy might lead to adverse pregnancy outcomes. However, another CgA-derived peptide, CST, seems to play a positive role in the modulation of the process of macrophage polarization toward an anti-inflammatory phenotype [6]. Similar, CST was also observed as a negative regulator of TNF-α [48,64]. This factor is a strong activator of nuclear factor kappa B (i.e., NFĸB),and is perceived as a main reason for pregnancy complications. Although high NFĸBactivity at the beginning of gestation is desirable to regulate the process of invasion and uterine spiral vessel transformation, the level of this factor should be silenced over the course of gestation to support maternal immunosuppression. In pathological pregnancies, especially in PE, both the level and activity of NFĸB are elevated, and this might be the reason that CST levels are depleted in pregnancies complicated by hypertension [51,65]. Unfortunately, although the impact of these CgA-derived peptides on immunological cells has been documented in numerous studies, at present, the mechanisms of CgA’s action on the molecular pathways regulating the activity of immunological cells have not yet been determined.

### 4.4. Apoptosis: Trophoblast Differentiation and Vascular Remodeling

Another process important for the successful course of pregnancy is programmed cell death. As mentioned previously, apoptosis is one of the processes involved in the remodeling of spiral arteries. Furthermore, it is also crucial for trophoblast formation. Although normally, it is involved in almost all stages of trophoblast differentiation and development, when uncontrolled, excessive trophoblast apoptosis may result in pregnancy complications [66]. Experiments with a use of chromogranin A and its derived peptides attempting to assess their role in apoptosis regulation have shown disparate results.

Studies performed on prostate cancer cells revealed a protective effect of CgA on the tested cells. The effect was dependent on up-regulation of the anti-apoptotic protein survivin through the Akt pathway [5]. Another experiment, conducted with CgA-specific antibodies, resulted in programmed cell death, confirming the anti-apoptotic role of the CgA protein [67] Furthermore, experiments conducted on AtT-20 pituitary cells and cultured rat cerebral cortical neurons revealed an anti-apoptotic effect of pGlu-serpinin, one of the serpinin peptides derived from CgA, which is also in accordance with previously described findings [31]. Contrary to those outcomes are the results of the studies performed on microglial cell cultures, where CgA exhibited a pro-apoptotic effect [68,69]. The observed differences may result from distinct derivations of tested cell lines and, once again, may point to local differences in CgA processing and the methods of action.Therefore, keeping in mind that CgA is present in the placenta, further studies are required to establish the precise effect of this hormone.

## 5. Chromogranin A as a Player in Cell Signaling Pathways of Pregnancy

The success of an implantation depends on extensive cross-talk between the endometrium, the trophoblast cells, and factors involved in the invasion and migration of the trophoblastic cells, as well as syncytia formation. Any irregularity may result in pregnancy complications such as intrauterine growth restriction, PE, or implantation failure. Therefore, the entire process is subjected to strict control by numerous autocrine and paracrine factors. Subsequently, each factor generates a signal, which follows a pathway that recruitsthe next molecules and gathers a linear or branched cascade, finally leading to the expression of effector molecules. 

CgA, besides being involved in various aforementioned pregnancy-related processes, shares some of the most important cell signaling pathways with molecules responsible for the pregnancy maintenance. These cell signaling pathways are involved in trophoblast invasion, migration, and syncythialization. Some of the most vital CgAs, and their derived peptides, cross-talk with pregnancy-related signaling trails visualized in Figure 3. 

### 5.1. Mitogen-Activated Protein Kinases (MAPKs) Signaling Pathway

In the MAPK signaling pathway, receptor tyrosine kinases (RTKs) are activated by the ligand. This process leads to the tyrosine residues’phosphorylation and activation of other signaling molecules, such as SHP2 (Src homology 2 (SH2)-domain containing tyrosine phosphatases) or Grb-2 (growth factor receptor-bound protein 2). Subsequently, Ras and Raf GTPases are activated, leading to the activation of MAPKs families, including ERKs (extracellular signal-regulated kinases), JNKs (c-Jun N-terminal kinases), and p38 MAPKs (Figure 3) [70].

Among these, the ERKs family is reported to play an important role in the trophoblast invasion and migration. It can be activated by various growth factors and cytokines; for instance, ERK ½ is activated by insulin-like growth factor-II (IGF-II) and insulin-like growth factor binding protein-I (IGFBP-I), both with a great impact on pregnancy outcome. Some data also suggest that there is cross-talk between the signal transducer and activator of transcription 3 (STAT3) and ERK ½ activation; therefore, both of these pathways are relevant for the invasion of trophoblast cells. Furthermore, ERK ½, as well as p38MAPK, are reported to be involved in trophoblast fusion and differentiation [70]. Not only trophoblast-related processes are connected with MAPK signaling pathways: activation of ERK ½ was shown to be important for cardiac growth during pregnancy [71], and p38 MAPK inhibition was shown to exert a positive effect on renal injury in pregnant rats [72]. Preventing p38 MAPK from activation has been additionally suggested as a potential target for therapeutic interventions to prevent adverse pregnancy outcomes mediated by stress factors [73] as well as to alleviate PE [74]. 

IGF-I activates the MAPKs signaling pathway, leading to the activation of the nuclear transcription factor cAMP response element-binding protein (CREB). According to Subbiah Pugazhenthi et al., CgA, being a CRE-dependent gene, can be activated by IGF-I through at least three pathways: PI 3-kinase, MEK/ERK, and p38/MAPK.It has been demonstrated that IGF-I has an stimulating impact on both CgA promoters and CgA mRNAs [75]. Furthermore, an additional experiment provided data according to which constitutively active Ras and Raf-1 (C-Raf) led to an increase in basal CgA activity in PC12 (rat pheochromocytoma) cells. On the contrary, studies conducted on BON cells (neuroendocrine pancreatic tumor cell line) imply that Raf-1 protein activation suppressed the expression of CgA [76]. At the same time, in another study conducted on BON cells, IGF-1 was shown to activate ADP-ribosylation factor (Arf). After the inhibition of Arfactivity or expression, secretion of CgA was significantly impaired [77].The presented differences in the obtained results may emerge from local differences in CgA function and cleavage (PC12 vs. BON cells) and/or may be explained by the existence of another signaling pathway cross-talking with the MAPKs family. Worth noticing is the fact that Raf-1 can activate not only ERK, but also nuclear factor-kappa B (NFkB), which is known to be involved in pregnancy-related processes [78]. Therefore, future studies in this area will provide additional data and allow us to understand the IGF-1-Raf-1-CgA relationship and its influence on pregnancy outcomes. 

### 5.2. Phosphoinositide 3-Kinase (PI3K)/AKT Signaling Pathway

Activation of G-protein-coupled receptors (GPCRs) or RTKs leads to the recruitment of p85 and p110 PI3K subunits to the membrane. Subsequently, phosphatidylinositol-4 and 5-bis-phosphate (PIP2) are phosphorylated and converted to PIP3. The above process results in the activation of AKT by phosphoinositide-dependent kinase-1 (PDK1), which further acts on downstream targets (Figure 3) [70]. The PI3K/Akt pathway has various functions in pregnancy; among others, it regulates decidualization and implantationas well as cellular processes, including cell growth, proliferation, migration, and survival [70,79,80].

One of the CgA-derived peptides, CST, was shown to activate the PI3K/Akt pathway (as well as ERK1/2), inhibiting endoplasmic reticulum stress-induced cell apoptosis [81]. A similar effect exerts CgA itself. In the study conducted by Junyang Gong et al. CgA was shown to increase PC cell survival through Akt-mediated survivin up-regulation [5]. The importance of PI3K/Akt signaling in CgA regulation is stressed by the fact that after PI3K inhibition and, therefore, reduced phosphorylated Akt, CgA expression is significantly suppressed [82]. As apoptosis is an essential feature of normal placental development, and is also associated with placental diseases [83], CgA’s influence on cell survival may have a great impact on this part of the regulatory system in pregnancy.

### 5.3. JAK-STAT Signaling Pathway

In the Janus kinase/signal transducer and activator of transcription (JAK/STAT) signaling pathway, the receptor binds with its cellular ligand, causing receptor dimerization and cross-phosphorylation of JAKs and cytoplasmic domains of the receptor. Subsequently, other domains and proteins possessing SH2-like STATs can be bound and phosphorylated on the specific tyrosine and serine residues. Afterwards, activated STATs dissociate and translocate into the nucleus, where they bind to specific promoters of the targeted genes [70]. The JAK/STAT signaling plays an essential role in embryogenesis and trophoblastic cell proliferation and invasion. 

As mentioned previously, CSTs not only activateERK1/2 and PI3K/Akt pathways to inhibit endoplasmic reticulum stress-induced cell apoptosis, but according to the Nour Eissa et al., in inflammatory conditions, CST may also activate the STAT3 pathway, inducing an anti-inflammatory effect and influencing cell proliferation and migration (Figure 3) [84]. Both apoptosis and inflammation, are key components of pregnancy homeostasis, and when present in excessive amounts, they can lead to pregnancy complications such as PE [85].

## 6. Conclusions

Pregnancy is a state of extreme physiological rearrangements and hormonal fluctuations. All observed changes are directly related to the needs of the developing fetus. One of the endocrine factors of pregnancy is CgA, an acidic protein produced, among others, by the placenta. Although CgA has not been properly studied in terms of pregnancy establishment and maintenance, the existing literature allows us to assume its possible influence on the course of pregnancy. This is evidenced by the involvement of CgA in cell signaling pathways related to trophoblast cell migration and invasion, as well as syncytia formation, and most importantly by the proven impact of CgA and its derived peptides on the regulation of angiogenesis, vascular remodeling, BP, and immune responses. All of the aforementioned processes are key compounds of the pregnancy adaptive system, although we do not yet fully understand the role of CgA in the complicated process of pregnancy, and to establish the exact mechanism of action of its cleavage products, further studies are necessary. While undertaking new scientific approaches regarding CgA’s role in pregnancy, special attention should be paid to the CgA-derived peptide catestatin. Further, emphasis should be placed on in vitro studies with the use of trophoblast cell cultures, reflecting placental conditions as thoroughly as possible.

## Figures and Tables

**Figure 1 ijms-24-04986-f001:**
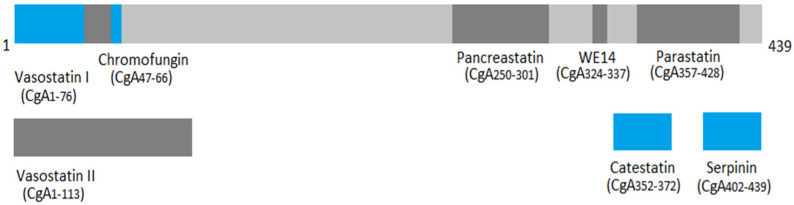
Chromogranin A and its most important biologically active peptides [6,21]. The blue color is reserved for the peptides with the highest significance regarding the discussed issue; light grey represents the whole CgA protein; dark gray rest of the marked peptides.

**Figure 2 ijms-24-04986-f002:**
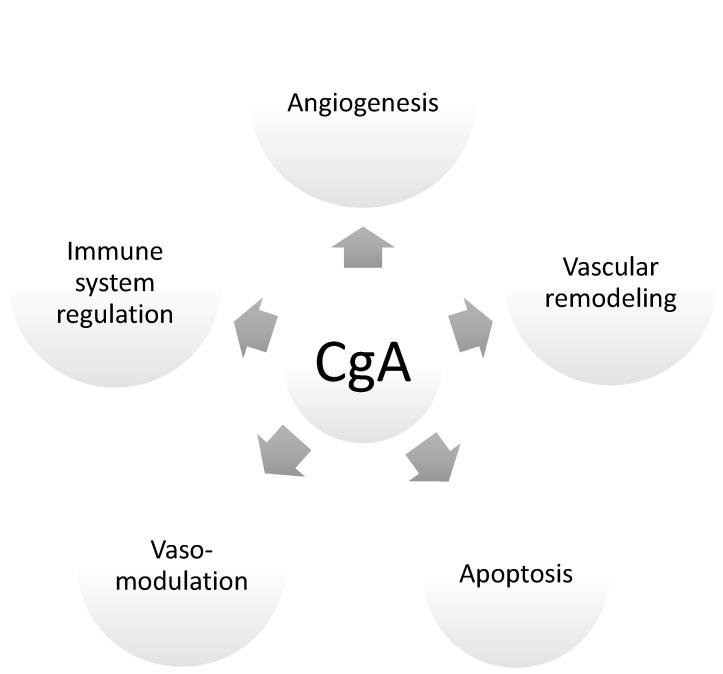
Processes of pregnancy regulated by chromogranin A (CgA).

**Figure 3 ijms-24-04986-f003:**
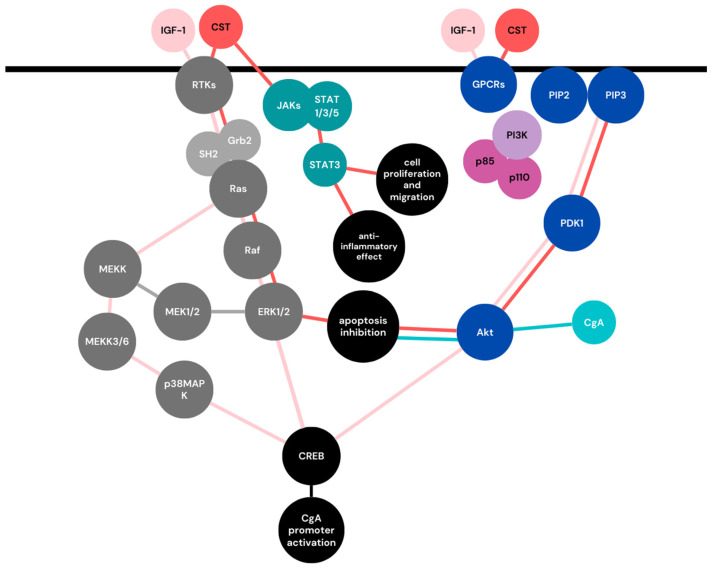
Pregnancy-related signaling pathways of chromogranin A and catestatin. Schematic representation of the signaling pathways activated duringpregnancy. MEKK: MAPK/ERK kinase. Light pink color marks pathways activated with IGF-1, resulting in CgA promoter activation. Orange color marks signaling pathways activated by CgA-derived peptide, CST. Additional relationship between CgA and Akt, through up-regulation of survivin protein, was marked with sea color. Gray color represents MAPKs signaling pathway, blue PI3K/Akt signaling pathway; black is reserved for each pathway’s end-effect related to CgA.

## Data Availability

Not applicable.

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
