# Peer review of "Chromogranin A: An Endocrine Factor of Pregnancy"

_ijms, 2023, doi:10.3390/ijms24054986_

Round 1
Reviewer 1 Report
This article has been gathering the current knowledge of chromogranin A function to gestation and parturition which includes the crucial pregnancy adaptations for CgA and CgA- derived peptides and CgA’s role in cell signaling pathways of pregnancy. The authors have included much detail about the possible affection for CgA with proper references. Here are several suggestions for this review:
1. In the ‘Introduction’ section, only mentioned there might be some significant role for CgA in pregnancy, but there is no preview for which function CgA could have for gestation and parturition. It would be easier to catch the main points for this article if there is short description here. Also table 1 is described in this section, since the next section is talking about the ‘Hormonal balance of pregnancy’, is this table more proper if shows in section 2?
2. In section 3, sentence ‘Granins belong to the family of acidic proteins produced by endocrine,… the most widely studied granin is chromogranin A (CgA )’ need cite proper references.
3. Section 5.1 has been talked about the possible affection for cardiovascular adaptations in pregnancy. Does this affection exist for people not in pregnant situation? Is there any comparison for CgA to normal people/ pregnancy for cardiovascular adaptations?
4. After the exploration of all possible functions for CgA to gestation and parturition, a discussion of further study directions for CgA is more helpful for audience to understand the significance of this work, also helpful for other scientists’ further studies.
5. Minor changes: 1) some acronyms are never mentioned (CD8+, CD8+T) 2) it’s easier to understand the context if figure 2 is put at the beginning of section 4
Reviewer 2 Report
Chromogranin A (CgA) is a protein widely distributed in various endocrine and neuroendocrine tissues and was identified in the placenta in 1996. Gathering and illustrating its function would contribute to the study of reproduction, especially pregnancy. However, there are some leading suggestions for this review manuscript.
- The information about CgA and its derived peptides needed to be better introduced. A table with detailed info for each of them is required.
- Since CgA is considering a tumor marker in clinical, the authors need to mention this and explain the potential relationship between the tumor and reproduction healthy.
- Overall, the link between CgA and pregnancy was weak. Although only a few publications have reported it, the logical structuring of this manuscript needs to change to strengthen the connection between CgA with pregnancy. I would suggest mentioning the direct function mechanisms of CgA and its peptides, then linking them to the main effectors (cytokines, signaling pathways, etc.) of pregnancy. In addition, any epidemiologic studies findings can be used?
- The author mentioned that only one report is attributed directly to CgA's role in pregnancy. Please highlight that paper and describe its work, limitations, and significance.
- The authors highlighted the immune function of CD8+ cells. However, the links between ChgA and CD8+ cells are weak. More info is needed, such as how ChgA activates the specific population of CD8+ cells (the mechanisms), The dynamic profile of CD8+ cells and its downstream cytokines during pregnancy, especially their profile in the placenta (any model used), and effects on placentation, and pregnancy maintenance?
- Any epi study can be used to stretch the link between CgA and Pregnancy?
- There are some misleading statements in the text, as listed below,
a. Line 7 & 373, "One of the endocrine factors of pregnancy is chromogranin A, an acidic protein produced by the placenta. Although", the CgA is produced in endocrine and neuroendocrine cells/tissues, not only by the placenta.
b. Line 269, Chromogranin A is not a tissue-specific hormone.
The minor issues that needed to be fixed are listed below:
- Table 1 needs to be more organized and ensure the "role in pregnancy" column aligns to the left.
- Lines 40-41, "Oestrogens are the group of steroid hormones, where the most abundant one is oestrogen" specify the Oestrogens and Oestrogen.
- Lines 72-73, "exhibit completely opposite responses." what do you mean?
- Lines 80-81, specify whether the effect is positive or negative.
- Line 234, needs to specify the regulation function.
- Lines 291-295, are those primary texts or the legend of Figure 3?
- Lines 321-322, what kind of impact?
- There are too many spacing issues throughout the manuscript.
- Since the full name has been introduced at the beginning of the manuscript, no full names are needed late on.
Round 2
Reviewer 2 Report
Need References for Lines 25-27;
Need References for Lines 27-28;
Need space between CD8+ and T for Lines276-304;
Delete ] in Line 344
Author Response
1.Need References for Lines 25-27; 2.Need References for Lines 27-28;
Answear: The following references were added:
- Jiaur G, Gayen R, Zhang K, et al Role of Reactive Oxygen Species in Hyper-Adrenergic Hypertension: Biochemical, Physiological, and Pharmacological Evidence from Targeted Ablation of the Chromogranin A (Chga). https://doi.org/10.1161/CIRCGENETICS.109.924050
- Gong J, Lee J, Akio H, Schlegel PN, Shen R (2015) Attenuation of Apoptosis by Chromogranin A-Induced Akt and Survivin Pathways in Prostate Cancer Cells. 148:4489–4499
- Muntjewerff EM, Dunkel G, Nicolasen MJT, Mahata SK, Van Den Bogaart G (2018) Catestatin as a target for treatment of inflammatory diseases. Front Immunol. https://doi.org/10.3389/fimmu.2018.0219
- Bílek R, Vlček P, Šafařík L, Michalský D, Novák K, Dušková J, Václavíková E, Widimský J, Zelinka T (2019) Chromogranin a in the laboratory diagnosis of pheochromocytoma and paraganglioma. Cancers (Basel). https://doi.org/10.3390/cancers11040586
3.Need space between CD8+ and T for Lines276-304;
Spaces were added where appropriate.
4.Delete ] in Line 344
The ] sign was delated.
The authors want to thank the Rewiever for each suggestion.